# A Systematic Review and Meta-Analysis of Symptoms of Anxiety, Depression, and Insomnia in Spain in the COVID-19 Crisis

**DOI:** 10.3390/ijerph19021018

**Published:** 2022-01-17

**Authors:** Stephen X. Zhang, Richard Z. Chen, Wen Xu, Allen Yin, Rebecca Kechen Dong, Bryan Z. Chen, Andrew Yilong Delios, Saylor Miller, Roger S. McIntyre, Wenping Ye, Xue Wan

**Affiliations:** 1Faculty of Professions, University of Adelaide, Adelaide, SA 5005, Australia; chenzbryan@gmail.com; 2Crescent Valley High School, Corvallis, OR 97330, USA; richardziychen@gmail.com; 3Nottingham University Business School China, University of Nottingham Ningbo China, Ningbo 315100, China; wen.xu@nottingham.edu.cn; 4School of Humanities, Southeast University, Nanjing 211189, China; allen-yin@hotmail.com; 5Business School, University of South Australia, Adelaide, SA 5001, Australia; Rebecca.dong@unisa.edu.au; 6Department of Psychology, University of Adelaide, Adelaide, SA 5001, Australia; delios9580@gmail.com; 7College of Business, Oregon State University, Corvallis, OR 97330, USA; millesay@oregonstate.edu; 8Mood Symptoms Psychopharmacology Unit (MDPU), Department of Psychiatry, University of Toronto, Toronto, ON M5T 2S8, Canada; Roger.McIntyre@uhn.ca; 9Department of Business Administration, School of Management, Jinan University, Guangzhou 510632, China; yzq721@yeah.net; 10School of Economics and Management, Tongji University, Shanghai 200092, China; wanxue@tongji.edu.cn

**Keywords:** general population, frontline healthcare workers, anxiety, depression, meta-analysis

## Abstract

Background: General population, frontline healthcare workers (HCWs), and adult students in Spain are at risk of anxiety, depression, and insomnia symptoms during the COVID-19 crisis. A meta-analysis of the individual studies on these symptoms would provide systematic evidence to aid policymakers and researchers in focusing on prevalence, risk, and best interventions. Objective: This paper aims to be the first meta-analysis and systematic review to calculate the prevalence of anxiety, depression, and insomnia symptoms in Spain’s adult population (general population, frontline healthcare workers (HCWs), and adult students) during the Covid-19 epidemic. Method: Random-effect meta-analysis was used to estimate the prevalence of anxiety, depression, and insomnia. Results: The meta-analysis includes 28 studies with 38 individual samples in Spain. The pooled prevalence of anxiety symptoms in 22 studies comprising a sample population of 82,024 was 20% (95% CI: 15–25%), that of depression symptoms in 22 articles with a total sample comprising 82,890 individuals was 22% (95% CI: 18–28%), and that of insomnia symptoms in three articles with a sample population of 745 was 57% (95% CI: 48–66%. Conclusions: The accumulative evidence reveals that adults in Spain suffered higher prevalence rates of mental symptoms during the COVID-19 crisis, with a significantly higher rate relative to other countries such as China. Our synthesis also reveals a relative lack of studies on frontline and general HCWs in Spain.

## 1. Introduction

Since COVID-19 was first reported on 17 November 2019, Spain, with its first case on 31 January 2020, is one of the countries in Europe most affected by infections, complications, and deaths. The COVID-19 crisis in Spain is particularly severe as a quarter of the population have been unable to isolate themselves according to public health recommendations [1]. Like most parts of the world, the COVID-19 pandemic has severely affected the Spanish population and has significantly altered the workplace and day-to-day activities. Disruptions such as social distancing laws, mandatory lockdowns, and a high risk of infection can lead to psychological problems [2]. A survey study of 17 countries by Luo et al. [3] found that Spain had one of the highest prevalence rates of anxiety during the COVID-19 pandemic.

The prevalence of mental health disorders has become an important research topic globally with differences across regions [4]. For example, individual studies have reported the prevalence of mild anxiety symptoms in Spain range from 1.8% [5] to 58.6% [6], while mild depression symptoms have varied from 7.3% [5] to 46% [6]. Given the disparity, it is especially pertinent to conduct a meta-analysis to pool the prevalence rates from individual empirical studies to generate accumulative meta-analytical evidence. For instance, Chen et al. (2021) conducted a meta-analysis on the prevalence of mental health during the COVID-19 crisis in Africa and identified the extent and pattern of prevalence of mental health symptoms that carry crucial implications and impact future research, in order to enable evidence-based medicine in Africa [7]. Pappa et al. (2021) performed a systematic review and meta-analysis of mental health symptoms during the COVID-19 pandemic in Southeast Asia and found a significant proportion experienced at least mild to moderate levels of anxiety and depression, in order to inform targeted identification of mental health symptoms and facilitate appropriate resource planning and allocation in that region [8]. The meta-analytic evidence of depression and anxiety mental health in Eastern Europe during the COVID-19 pandemic helps to reveal the high prevalence rates of clinically significant symptoms [9] and calls for more effort to conduct such meta-analysis studies to assist evidence-based mental healthcare in other regions. 

Answering this call and following such studies in other regions, this study aims to provide meta-analytical evidence on the prevalence rate of mental health symptoms in Spain to help with evidence-based decisions on the allocation of limited resources during the continuing COVID-19 crisis. This study conducts the first systematic review and meta-analysis of mental symptoms in Spain’s adult population by assessing the prevalence of anxiety, depression, and insomnia symptoms during the COVID-19 virus. The adult populations include the key populations that have been studied by individual papers, including healthcare workers, general adult population, and adult (university) students [10]. This systematic review and meta-analysis of empirical studies will help to guide mental health practitioners, policymakers, and researchers in their effort to better understand and address mental health in Spain.

## 2. Methods and Materials

We registered with the International Prospective Register of Systematic Reviews (PROSPERO: CRD42020224458) and used the Preferred Reporting Items for Systematic Reviews and Meta-Analyses (PRISMA) statement 2020 to guide our search [11]. 

### 2.1. Data Sources and Search Strategy

This paper comes from a large project—a meta-analysis of mental health symptoms during COVID-19. We searched the following databases for studies that fit our criteria: PubMed, Embase, PsycINFO, and Web of Science from 1 February 2020 to 6 February 2021. To identify articles meeting the requirements, we searched specific titles and abstracts using the keywords in Appendix A with Boolean operators. We also searched the preprints at medRxiv.

### 2.2. Selection Criteria

The search included empirical studies that reported the prevalence of anxiety, depression, or insomnia symptoms among frontline HCWs, general HCWs, general adult populations, or adult (university) students in Spain. To be included, reports had to implement validated psychometric measures with outcomes reported in English.

We excluded studies that had populations of children, adolescents, or adult subpopulations (e.g., pregnant women). We also excluded non-original research or studies that were reviews, meta-analyses, qualitative and case studies, interviews, news reports, interventional studies, or articles without validated instruments or validated cutoff scores to identify prevalence.

A researcher (WX) contacted the authors of papers that lacked important information in several instances: (1) if they surveyed a population that included both targeted and excluded populations in such a way that we could not identify the prevalence rate for our desired population; (2) if the paper included primary data meeting our inclusion criteria, but did not report the prevalence; (3) if the paper reported the overall prevalence without specifying its cutoff point to determine whether it was mild above or moderate above; or (4) if the paper was missing, or unclear about, critical information such as respondent rate, data collection time, or female proportion rate. 

### 2.3. Data Screening 

To begin, we exported article information from various databases into Endnotes to remove duplicates and then imported them into Rayyan. Two researchers (BZC & AD) independently screened the titles and abstracts of all papers using our inclusion and exclusion criteria. Any conflicts were resolved by a third researcher (RKD). 

### 2.4. Data Extraction

A well-developed coding protocol and coding book were developed based on previous studies [12]. All included articles from the screening were sent to two pairs of researchers who were assigned to thoroughly examine and extract important data into a coding book (WX & AY, BZC & AYD, RZC & SM). Using a coding procedure, the coders coded relevant information including author, title, country, start and end dates of data collection, study design, population, sample size, respondent rate, female proportion rate, age range and mean, outcome, outcome level, instruments, cutoff scores, and prevalence. We coded the severity prevalence as mild above, moderate above, and severe, if possible. Studies that reported mild, moderate, and severe prevalence were converted into mild above, moderate above, and severe prevalence. For studies that only reported the overall prevalence, we specified their severity if their cutoff points were reported.

Comments related to reasons for emailing and excluding papers were recorded. After both coders had independently coded their articles, they cross-checked their information. Different answers were conferred about and changed, and disagreements were settled by a third coder coding the article. The lead coder (RZC) double-checked important data including the population, sample size, mental health outcomes, outcome levels, instruments, and prevalence. Papers with unusual prevalence, cutoff scores, and numbers were double-checked for sensitivity analysis.

### 2.5. Assessment of Bias Risk

The Mixed Methods Appraisal Tool (MMAT) [13], a seven-question test that assesses the quality of research papers, was used for our meta-analysis. Pairs of coders individually coded these questions. Discrepancies were resolved through the lead coder (RZD). The quality scores of papers ranged from 0 to 7. Studies with a quality of above 6 were considered high, articles with a score of between 5 and 6 were classed as medium, and articles with a score of below 5 were considered low.

### 2.6. Data Analysis 

Using Version 16.1 of Stata (Stata Corp LP, College Station, TX, USA), we conducted a random-effect meta-analysis to calculate the pooled prevalence from multiple studies using meta-prop. For each mental health symptom, we also conducted subgroup analyses by population (frontline HCWs, general population, and adult students), severity (above mild, above moderate, severe, and overall), and popular instrument (DASS-21 vs. GAD-7/PHQ-9). 

### 2.7. Sensitivity Analysis 

Research methodologists have found that the conventional funnel plots to assess biases in meta-analyses are inaccurate for proportion studies [14]. In meta-analysis of proportion studies, which was our approach, a Doi plot and the Luis Furuya-Kanamori (LFK) index represent the better approach for graphically representing publication bias—where a symmetrical triangle implies the absence of publication bias, while an asymmetrical triangle indicates possible publication bias [15]. The Doi plot and LFK index have higher sensitivity and power to detect publication bias than the funnel plot and Egger’s regression [16].The LFK index provides a quantitative measure to assess the degree of asymmetry—a score within ±1 indicates ‘no asymmetry’, exceeds ±1 but is within ±2 indicates ‘minor asymmetry’, and exceeds ±2 indicates ‘major asymmetry’.

## 3. Results 

### 3.1. Study Screening

Figure 1 illustrates the PRISMA flow chart for our search and data extraction process. We found 6949 citations from the selected database and other sources. After excluding 2729 entries that did not meet the inclusion criteria in the screening process, we extracted data from 684 citations based on their full text. After excluding 505 citations in the data extraction process, we coded 150 papers that included the necessary information to conduct the meta-analysis. In addition, we sent out two rounds of emails to the authors of 95 papers to request useful information. We received 75 responses. Among these responses, we received new prevalence data from eight out of the 29 studies that lacked prevalence data. Finally, we had 168 studies providing necessary information for meta-analysis, among which 28 articles had participants that were from Spain.

### 3.2. Study Characteristics 

In total, this meta-analysis includes 38 samples from 28 studies with 86,323 participants from Spain [6,17,18,19,20,21,22,23,24,25,26,27,28,29,30,31,32,33,34,35,36,37,38,39,40,41,42,43,44], as summarized in Table 1. These articles were noted with an asterisk in the reference list. Among them, 30 samples (78.95%) were of general populations, three samples were of frontline HCWs (7.89%), and five (13.16%) were of adult students. Most studies were cross-sectional (85.71%), and four studies were longitudinal cohort studies (14.29%). Almost all studies were published (92.86%), while only 2 (7.14%) remained as preprints. The sample size of the 38 samples varies from 44 to 21,207, with a medium value of 1199. The participation rates varied from 20% to 98%, with a medium value of 70.3%. The female proportions among the 38 samples varied from 0% to 100% with a medium value of 70.25%. Appendix A reports the characteristics of each study. 

### 3.3. Pooled Prevalence of Anxiety, Depression, and Insomnia

In Spain, 29 samples from 22 studies reported the prevalence of anxiety symptoms among 82,024 participants. The most common anxiety instruments used included the Depression, Anxiety and Stress Scale—21 Items (DASS-21) (54.5%) and the Generalized Anxiety Symptoms 7-items scale (GAD-7) (31.8%). Different studies used different cutoff values to determine the overall prevalence, as well as the severity of anxiety. In the random-effects model, the pooled prevalence of anxiety was 20% (95% CI: 15–25%) in the 22 studies (Figure 2A), indicating in general that 20% of the adults in Spain had anxiety symptoms during the COVID-19 pandemic. Based on a normal distribution, the prediction internal is 0–67%, which is the range of the prevalence of anxiety symptoms in comparable studies.

Figure legend: The square markers indicate the prevalence of the symptom at the different level for different populations. The size of the marker correlates to the inverse variance of the effect estimate and indicates the weight of the study. The diamond data marker indicates the pooled prevalence.

A total of 20 samples from a total of 22 articles that we reported in this meta-analysis were on depression, for a total of 82,890 respondents. The most commonly included depression instruments were used, with DASS-21 being the most popular (50%) along with Patient Health Questionnaire (PHQ)-9 (31.8%). In the random-effects model, the pooled prevalence of depression was 22% (95% CI: 18–28%) among the 22 studies (Figure 2B). Its prediction internal is 0–65% and the prevalence of insomnia symptoms in comparable studies will fall within this range.

Five samples of the three articles that we reported in this meta-analysis reported on insomnia, for a total of 745 respondents. The Insomnia Severity Index (ISI) and Pittsburgh Sleep Quality Index (PSQI) were used to measure insomnia. In the random-effects model, the pooled prevalence of insomnia is 57% (95% CI: 48–66%) (Figure 2C). Its prediction internal is 25–87%.

The overall prevalence of mental health symptoms in Spain’s frontline HCWs, students, and general population are 50%, 42%, and 19%, respectively. The overall prevalence rates of mental health symptoms that surpassed the cutoff values of mild, moderate, and severe symptoms were 38%, 18%, and 7%, respectively (Table 2). I^2^ vary from 83.5% to 99.4%, similar to those in the published meta-analyses on mental health during COVID-19 [3,45,46,47,48,49,50].

### 3.4. Quality of Articles

Using the Mixed Methods Appraisal Tool (MMAT)^18^, we found that seven studies (25%) are of higher quality, 21 studies (75%) had medium quality, and 0 studies were of low quality (Table 1). The subgroup analysis suggests the studies with high quality reported lower prevalence of mental health symptoms in Spain (Table 2).

### 3.5. Sensitivity Analysis 

Figure 3 depicts the Doi plot and a Luis Furuya–Kanamori (LFK) index of −0.45, indicating ‘no asymmetry’ and the unlikely presence of publication bias. We also tested the impact of publication status and sample size and did not find significant influence.

## 4. Discussion

### 4.1. Overview of the Findings

To date, this meta-analysis is the first to report the pooled prevalence of mental health symptoms in key populations in Spain during the COVID-19 crisis. Our study included data from 38 samples and 28 articles for a total of 86,323 adults in a single year of the COVID-19 pandemic. The pooled prevalence rates of anxiety, depression, and insomnia symptoms at the mild above level are 34%, 36%, and 52%, respectively (Table 3). The analysis found that both students and frontline HCWs had much higher prevalence rates of mental health symptoms than the general population. The foregoing finding of greater risk in students and frontline HCWs is in accordance with what has been reported elsewhere. The higher prevalence in students may be due to their tendencies to worry over the health both of themselves and loved ones as well as the pressures they face [51,52], while daily interactions in jobs with high-risk virus transmission could impact frontline HCWs [53].

Insomnia appeared to be much more prevalent than anxiety and depression in Spain, with 57% of the population being diagnosed with insomnia. Part of the reason for this might be that all five samples on the prevalence of insomnia used a mild above severity level, yielding a higher prevalence rate. 

### 4.2. Comparison with Prior Meta-Analyses 

Given that this meta-analysis is the first on the topic in Spain, it is worthwhile to compare the findings with similar meta-analyses in other countries. Most meta-analyses on COVID-19 mental health were conducted in a few geographical regions, with the majority in China, the first country to experience the COVID-19 crisis. The pooled prevalence of anxiety symptoms in Spain’s population was 34%, which is much higher than that reported in Bareeqa et al. (2020) (22%), Pappa et al. (2020) (23%), Krishnamoorthy et al. 2020 (26%) and Ren et al. 2020 (25%) [50,54,55,56]. Similarly, the pooled prevalence rate for depression in Spain’s population is 35%, which is much higher than reported in Bareeqa et al. (2020) (27%), Pappa et al. (2020) (23%), Krishnamoorthy et al. 2020 (26%), and Ren et al. 2020 (28%) [50,54,55,56]. The prevalence rates in Spain exceed the reported prevalence rates of mental health symptoms in China, suggesting that a higher percent of people in Spain suffered greater mental health symptoms. Differences across countries may be due to differences in social determinants and exposure to media [57].

In addition, the pooled prevalence of mild above anxiety in Spain is higher than those of a meta-analysis including data collected in 10 counties (China, India, Japan, Iran, Iraq, Italy, Nepal, Nigeria, Spain, and UK) (32%) [45] and is similar to those of another meta-analysis including data collected in 17 countries (i.e., China, Singapore, India, Japan, Pakistan, Vietnam, Iran, Israel, Italy, Spain, Turkey, Denmark Greece, Argentina, Brazil, Chile, and Mexico) (33%) [3]. The pooled prevalence for depression in Spain is also higher than those in a study of 17+ countries reported by Luo et al. [3] (28%) and higher than another study of 10 countries reported by Salari (34%) [45]; however, the prevalence rate for depression in Spain is still lower than Italy, which has the highest rate (67%) [3].

The pooled prevalence of anxiety during COVID-19 is much higher than the overall pooled prevalence rate of anxiety of people ages 65+ from nine studies in Spain before COVID-19 (11%, *n* = 12,577) [58]. 

The pooled prevalence rate for insomnia among Spain’s population (57%) is much higher than those in Germany (39%) and China (27%), as well as the pooled prevalence in 13 countries including Australia, Bahrain, Canada, Germany, Greece, Iraq, India, Mexico, and USA (36%), and is similar to Italy (55%) and France (51%), two countries with the highest prevalence rates [59]. 

The pooled prevalence rate of mental symptoms of the student population in Spain (50%) is higher than other meta-analyses, such as 28% on anxiety reported by Lasheras et al. [60] and 32% on anxiety, 36% on depression, and 33% on sleep disturbance reported by Deng et al. [51].

### 4.3. Practical Implications

The findings of this meta-analysis show that there were higher rates of anxiety, depression, and especially insomnia symptoms reported in Spain, compared to all the other countries reported apart from the nearby European countries of Italy and France. The evidence suggests the need to pay attention to the high prevalence rate of insomnia. As the literature shows that insomnia can be caused by fear of infection, quarantine, and death, required safety precautions such as masks and isolation methods can be critical [48,61]. It is also reported that extensive use of social media during COVID-19 is associated with higher rates of anxiety and insomnia-related symptoms.

The evidence from this study shows that frontline HCWs and students suffered more anxiety, depression, and insomnia symptoms than the general population, which suggests that policymakers and healthcare organizations need to prioritize frontline HCWs and students in this ongoing pandemic. Frontline HCWs are at a greater risk for mental health symptoms, which could provide the impetus for policies and strategic priorities towards building resiliency in HCWs. Protective measures, such as intercessions, self-help resources, and specified care of healthcare workers, are of the utmost importance [62]. An additional reason for prioritization is to preempt a possible rise in suicide associated with mental health issues during COVID-19 [63].

The finding that insomnia had the highest prevalence rate relative to anxiety and depression in Spain suggests a clarion call to probe populations for the presence of insomnia during the COVID-19 pandemic. It is also important to explore underlying causative factors resulting in the extraordinarily high rates. Moreover, our findings on the high rate of insomnia compared to anxiety and depression, which have been the most studied mental health symptoms, acts as a call for future research on insomnia, especially given that more than 90% of existing empirical studies studied anxiety and depression [64].

Lastly, to our surprise, no studies have examined the mental health symptoms of general HCWs in Spain. A possible reason for this could be that all general HCWs are on the frontline given the needs of the country. More research overall is needed to investigate the prevalence in HCWs.

### 4.4. Limitations and Future Research

Our meta-analysis had several limitations. Among the 38 samples included in this meta-analysis, 30 samples (78.95%) investigated the prevalence of the general population in Spain with only three samples on frontline HCWs and five samples on students. The lack of sufficient data from HCWs and students limited the reliability of the pooled prevalence rates.

Secondly, we found that articles used different instruments and inconsistent cutoff scores, which makes it difficult and often impossible to accumulate and compare research findings. For example, the prevalence of anxiety measured by GAD-7 (31%) and DASS-21 (15%), the two most popular scales, differed significantly in Spain (Table 3). Additionally, several articles reported an “overall” prevalence without indicating the cutoff points used, which makes it impossible to know which outcome level they used. We suggest that future research specifies the severity levels and the cutoff points used, especially to report overall prevalence. Third, non-English articles were not included and hence could create biases in our study.

## 5. Conclusions

This is the first meta-analysis to provide evidence on the pooled prevalence rates of anxiety, depression, and insomnia in Spain during the COVID-19 pandemic. Our results indicate that the people in Spain are experiencing psychological and emotional distress during COVID-19, possibly greater than in some other countries. Therefore, it is critical to identify the prevalence of specific psychological symptoms of the key populations to guide mental health assistance and resource allocation efforts in the unprecedented crisis.

## Figures and Tables

**Figure 1 ijerph-19-01018-f001:**
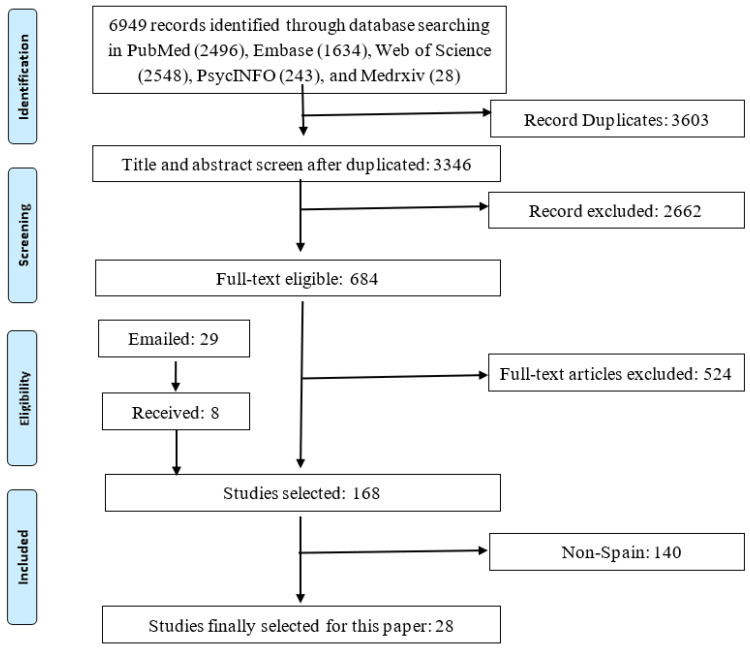
A PRISMA flow diagram.

**Figure 2 ijerph-19-01018-f002:**
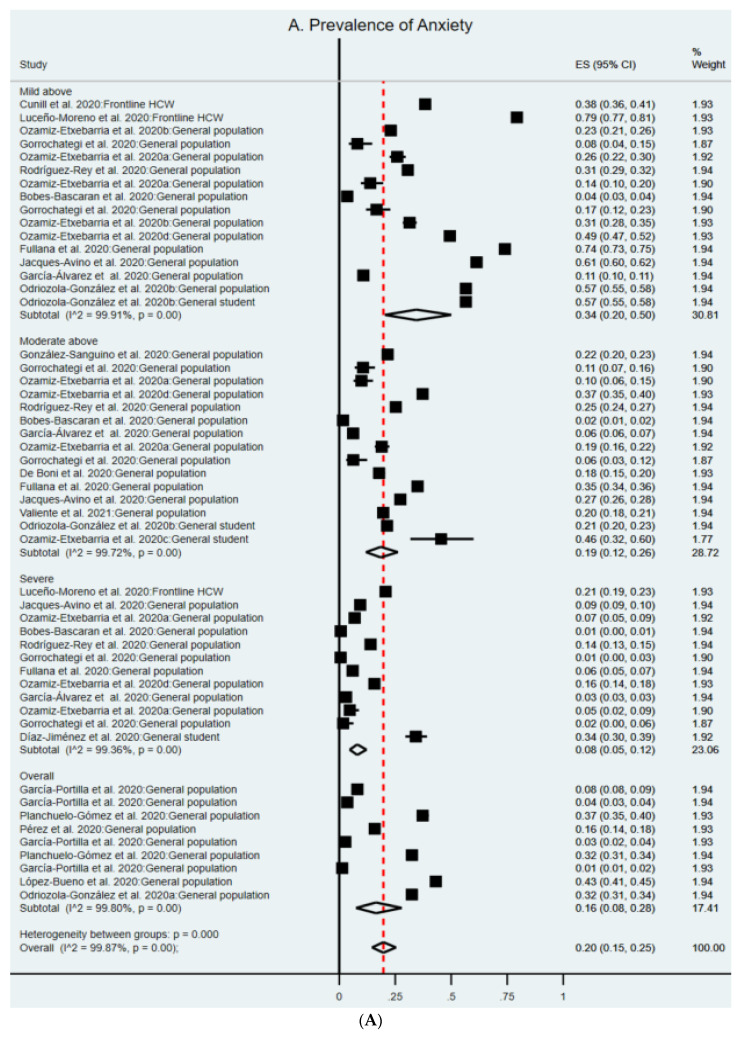
(**A**–**C**). Forest plot of the prevalence of anxiety, depression and insomnia.

**Figure 3 ijerph-19-01018-f003:**
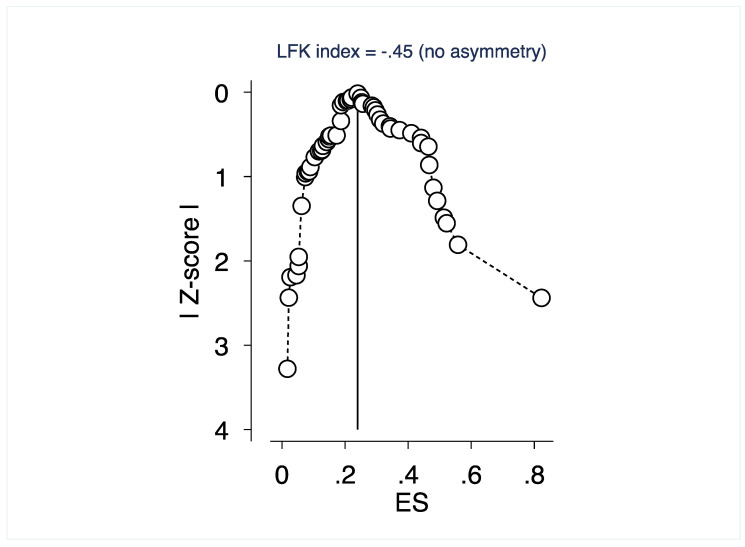
Depiction of publication bias in the baseline meta-analysis of proportion studies based on Doi plot and the Luis Furuya-Kanamori (LFK) index -a score that is within ±1 indicates ‘no asymmetry’.

**Table 1 ijerph-19-01018-t001:** Study characteristics for mental health symptoms in COVID-19 epidemic in Spain.

Characteristics	Total Number of Studies/Samples *	Percent (%)	Level of Analysis
Overall	28/38	100	
Design			Study
Cohort	4	14.29	
Cross-sectional	24	85.71	
Publication status			Study
Preprint	2	7.14	
Published	26	92.86	
Quality			Study
>6	7	25.0	
Between 5 and 6	21	75.0	
<5	0	0.0	
Population			Sample
Frontline HCW	3	7.89	
General population	30	78.95	
Student	5	13.16	
Outcome #			Prevalence
Anxiety	52	47.1	
Depression	52	47.1	
Insomnia	5	4.59	
Severity #			Prevalence
Above mild	39	35.78	
Above moderate	30	27.52	
Severe	23	21.1	
Overall	17	15.6	
	Median (mean)	Range	
Sample size	1199 (2272)	44–21207	Sample
Response rate	70.3% (73.9%)	20.0–98.0%	Sample
Female portion	70.25% (64.7%)	0–100%	Sample

* One study may include multiple independent samples such as frontline HCWs and general population [31]. # The total prevalence of mental health outcomes is larger than the 38 samples because one sample can assess multiple mental health outcomes including anxiety, depression, and insomnia. Similarly, a study may report multiple levels of severity on each mental health outcome for each sample.

**Table 2 ijerph-19-01018-t002:** The pooled prevalence rates of mental health symptoms by subgroups of population, outcome, and severity.

First-Level Subgroup	Second-Level Subgroup	Prevalence (%)	95% CI
	Aggregated prevalence	22%	18–26%
Population	Frontline HCW	42%	22–64%
	General population	19%	16–23%
	Student	50%	32–69%
Outcome	Anxiety	20%	15–25%
	Depression	22%	18–28%
	Insomnia	57%	48–66%
Severity	Above mild	38%	30–46%
	Above moderate	18%	14–21%
	Severe	7%	5–9%
	Overall	25%	16–34%
Quality	Studies with high quality	21%	15–27%
	Studies with medium quality	23%	19–27%

Note: CI = Confidence Interval.

**Table 3 ijerph-19-01018-t003:** Subgroup analyses of the prevalence of anxiety, depression, and insomnia.

Groups	Subgroups	Anxiety	Depression	Insomnia
	Number of studies	22	22	3
	Number of samples	29	30	5
	Total # participants	82,024	82,890	745
	Aggregated prevalence	20%, 95% CI: 15–25%	22%, 95% CI: 18–28%	57%, 95% CI: 48–66%
Population	Frontline HCW	46%, 95% CI: 14–80%	33%, 95% CI: 06–69%	57%, 95% CI: 47–66%
	General population	17%, 95% CI: 12–22%	20%, 95% CI: 16–25%	55%, 95% CI: 48–61%
	Student	39%, 95% CI: 18–62%	59%, 95% CI: 58–61%	64%, 95% CI: 59–68%
Severity	Above mild	34%, 95% CI: 20–50%	36%, 95% CI: 29–43%	57%, 95% CI: 48–66% *
	Above moderate	19%, 95% CI: 12–26%	17%, 95% CI: 14–20%	
	Severe	8%, 95% CI: 5–12%	6%, 95% CI: 4–8%	
	Overall	16%, 95% CI: 5–12%	35%, 95% CI: 28–43%	
Instrument	DASS-21	15%, 95% CI: 11–20%	21%, 95% CI: 15–28%	
	GAD-7/PHQ-9	31%, 95% CI: 17–46%	22%, 95% CI: 13–32%	

Note: 95% CI = 95% Confidence Interval. * For insomnia, all severity level is mild above in the studies we examined.

## Data Availability

All data are available through https://www.dropbox.com/sh/5068kumz1o71v5d/AAAKs149KGg_XJeR7wiGALO4a?dl=0 (accessed on 28 August 2021).

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
