# Peer review of "A Systematic Review and Meta-Analysis of Symptoms of Anxiety, Depression, and Insomnia in Spain in the COVID-19 Crisis"

_ijerph, 2022, doi:10.3390/ijerph19021018_

Round 1

Reviewer 1 Report

Thank you for doing this important work. Is a current topic of great importance. I hope that the following points will help you to strengthen this meta-analysis.

Abstract

Background section is not correct. It seems an objective instead of a real background.

Several grammatical English mistakes For example, objective “To be the first metaanalyses”, is meta-analysis.

“With 38 unique samples in Spain” not clear.

Punctuation errors, grammar in general

The standard of references, style and manuscript format of the journal were not followed.

Incorrect introduction, very simple and general.

Mixing samples of healthcare professionals and students does not allow for clear results. There is a great risk of bias in the results reported by the authors

The meta-analysis of prevalence  is totally wrong. The authors from where the data are extracted are not detailed. To carry out a correct meta-analysis I encourage the authors to follow: https://handbook-5-1.cochrane.org/chapter_9/9_4_4_3_random_effects_method.htm

Reviewer 2 Report

I think the authors did an excellent job to assess the prevalence of Anxiety, depression and insomnia symtpoms in Spanish population using the data that is avaialable. I reccomend the editor to publish this work with some minor revisions.

1. In the discussion section can authors be more specific on what they mean by the below points ie do they mean by pooled prevalence of anxiety in spain is similar or different in countries like china and india. As they mentioned it twice saying that the pooled prevalence is higher in spain and then immediately they say it is similar.  From line 266-269. 

Reviewer 3 Report

This is an interesting study mainly due to its methodological approach. I would recommend it to be published after the following changes

Title:

 "... Mental Health Symptoms ..." in the title is very general, I would ask the authors to replace it with "... symptoms of anxiety, depression and insomnia ...".

Abstracts:

-From line 22, delete "Background:"

On line 38 "Trial record: CRD4202224458" what does it mean?

Introduction:

    - I wonder if it is necessary to refer to numbers for the victims of the pandemic. The number of victims changes dramatically every day.

Data sources and search strategy

- Give more details about the big project. If there are similar works that have been published I would ask you to change the phraseology so that they do not overlap. It is a mistake made in large surveys that produce more than one article.

Selection Criteria elsewhere

-Please remove the researchers' initials.

Results

-Check the values ​​in the PRISMA table (the table looks like the table you published in another article may be wrong or it may be similar by chance).

-Add a table with the characteristics of the studies and the overall prevalence rates of the symptoms of anxiety, depression and insomnia.

3.3 The cumulative prevalence of anxiety, depression and insomnia

-In Figure 2 A, B, C from the study column the data of each study are missing, add them.

-Line 193 Replace the word stress with the word depression.

-Line 202 Replace the word stress with the word insomnia.

-Is paragraph 3.4 missing or was the numbering wrong?

Discussion

I would ask researchers to add a paragraph comparing their findings with the findings of work in Spain before COVID. 
